# Protection of Human Rights and Barriers for People with HIV/AIDS in Colombia: An Analysis of the Legal Framework

**DOI:** 10.3390/ijerph191811423

**Published:** 2022-09-10

**Authors:** Sandra M. Parra-Barrera, María del Mar Sánchez-Fuentes, Nieves Moyano, Reina Granados

**Affiliations:** 1Department of Criminal Law, Philosophy of Law and History of Law, University of Zaragoza, 50009 Zaragoza, Spain; 2Department of Social Sciences, Faculty of Human and Social Sciences, University of La Costa, Barranquilla 080002, Colombia; 3Department of Psychology and Sociology, Faculty of Human and Social Sciences, University of Zaragoza, 44003 Teruel, Spain; 4Faculty of Humanities and Science Education, University of Jaén, 23009 Jaén, Spain; 5Department of Nursing, Faculty of Health Sciences, University of Granada, 18071 Granada, Spain

**Keywords:** HIV/AIDS, legal framework, human rights, protection, barriers

## Abstract

People living with HIV/AIDS not only suffer in terms of physical and/or psychological health, but also frequently experience violations of human rights and fundamental freedoms. Although there are international treaties and a regulatory framework that legally protects people with HIV/AIDS, it is essential to determine the effectiveness of the regulatory framework in Colombia. Therefore, our main goal was to examine the legislation on HIV/AIDS in Colombia with the purpose of understanding the decrees and laws, and describing the main obstacles and barriers that people with HIV/AIDS encounter. For this purpose, we employed the method of legal interpretation and reviewed the legal regulations on HIV/AIDS, as well as the judgments of guardianship of the Constitutional Court. It is verified that there is a specific regulation on HIV/AIDS, specifically decree 559 of 19,991, decree 1543 of 1997, Law 599 of 2000, Law 972 of 2005, and Law 1220 of 2008. Although at the legislative level Colombia shows an evolution in the norm, patients with HIV/AIDS continue to be victims of human rights violations. As a result, and through the analysis of tutela judgments, it was found that the Constitutional Court recognized the violation of rights and ordered the necessary measures to be taken to guarantee the human rights and fundamental freedoms of the defendants.

## 1. Introduction

The human immunodeficiency virus (HIV) attacks the immune system and, if left untreated, can cause acquired immune deficiency syndrome (AIDS). Despite the fact that there is no cure, in the case that a person who contracts HIV will have it chronically, both adequate medical care and adherence to treatment are essential for people to have long, healthy lives, and to prevent the development of AIDS [1].

According to the most recent data from the Joint United Nations Program on HIV/AIDS, in Colombia in 2020, 180,000 people were living with HIV/AIDS [2]. Similar data were provided by the Ministry of Health, which reported 177,871 cases of HIV in 2020 [3]. However, there was a decrease in cases in 2021, probably due to the global pandemic, where 134,636 people were reported to be living with HIV [4]. Figures regarding the number of diagnosed individuals who receive treatment and achieve viral suppression vary by the reporting agency. According to UNAIDS, 68% of people living with HIV have been diagnosed [2], while the Ministry of Health [3] estimated a rate of 71.1%. The High Cost Account reported a rate of 94.63% [4]. Regarding the rate of diagnosis among people who receive treatment, UNAIDS estimated it as 64% [2], compared to 95.28% according to the High Cost Account [4]. Finally, regarding the rate of viral suppression, UNAIDS reported that 57% of people who received treatment had a suppressed viral load [2], while the High Cost Account reported that the rate varies between 66.5% (people with <50 copies of HIV per milliliter of blood), 73.4% (people with <200 copies of HIV per milliliter), and 77.1% (people with <1000 copies of HIV per milliliter) [4].

UNAIDS proposed the global goal of “90-90-90” for 2020, i.e., that 90% of people living with HIV should have been diagnosed, 90% of those diagnosed should receive treatment, and 90% of those receiving treatment should have achieved viral suppression [5]. However, according to data from UNAIDS [2], Colombia did not meet the 90-90-90 target. Meanwhile, however, according to data from the High Cost Account, Colombia did meet the related diagnostic and treatment targets [4].

As well as physical health problems, such as well-known opportunistic diseases [6] and psychopathological disorders [7,8], the human rights and fundamental freedoms of people with HIV/AIDS are often violated [9]. The latter is especially important, since violation or non-protection of human rights exacerbates the spread and impact of the disease [10].

Therefore, the legal framework for HIV/AIDS is supported by general international instruments pertaining to the protection and guarantee of human rights and fundamental freedoms. The international instruments that recognize and guarantee the fundamental rights of all people are as follows: The Universal Declaration of Human Rights of 1948 [11], International Convention on the Elimination of All Forms of Racial Discrimination of 1965 [12], International Covenant on Economic, Social and Cultural Rights of 1966 [13], International Covenant on Civil and Political Rights of 1966 [14], American Convention on Human Rights of 1969 [15], Convention on the Elimination of All Forms of Discrimination against Women of 1979 [16], Additional Protocol to the American Convention on Human Rights in the area of Economic, Social and Cultural Rights of 1988 [17], Convention on the Rights of the Child of 1989 [18], and Inter-American Convention to Prevent, Punish and Eradicate Violence against Women of 1994 [19].

Furthermore, the legal framework is supported by specific international guidelines for HIV/AIDS. Some of the most important guidelines, in chronological order, are as follows: The Declaration of Helsinki of the World Medical Association of 1964 [20], Program of Action of the International Conference on Population and Development of 1994 [21], Beijing Platform for Action of 1995 [22], Protocol for the Identification of Discrimination against People Living with HIV of 2000 [23], Millennium Declaration of 2000 [24], Declaration of Commitment in the Fight against 2001 HIV/AIDS [25], 2001 International Labor Organization Code of Practice on HIV/AIDS and the World of Work [26], 2002 Strategic Orientation on HIV Prevention [27], Resolution on access to medicines in the context of pandemics of 2004 [28], International Guidelines on Human Rights and HIV/AIDS of 2006 [9], and Consolidated recommendations for the prevention, care, and treatment of co-infection with hepatitis B and C viruses in people with HIV infection of 2019 [29].

In accordance with the various binding instruments, Colombia, among other countries, must respect, protect, and comply with the following. First, they must respect human rights norms, and thus must not directly or indirectly violate the fundamental rights of people living with HIV/AIDS. Second, they must protect and adopt the necessary measures to prevent the human rights and fundamental freedoms of people living with HIV/AIDS from violation violated. Third, they must comply with the established norms, and develop and implement all necessary legislative, financial, administrative, and judicial measures.

Despite the existence of international treaties and specific guidelines, no previous study has examined the effectiveness of the regulatory framework designed to legally protect people with HIV/AIDS in Colombia. Therefore, the main objective of this research is to examine the legislation on HIV/AIDS in Colombia. The specific objectives are to (1) analyze the decrees and laws on HIV/AIDS, and (2) describe the main obstacles faced by people with HIV/AIDS according to the tutela rulings of the Colombian Constitutional Court.

## 2. Materials and Methods

### 2.1. Study Design

We use the method of legal interpretation in this study, since it is the most appropriate method for analyzing and interpreting the law according to our particular objectives [30]. It is important to note that the law goes beyond the legal precepts to be applied, and instead can be characterized as an interpretative social practice; the method of legal interpretation is the most appropriate to reveal deficiencies in the current rules [30]. 

### 2.2. Materials

The materials in this study were decrees and laws pertaining specifically to HIV/AIDS in Colombia, in addition to the five most recent tutela judgments by the Constitutional Court. 

### 2.3. Procedure

First, we examined the laws that establish rights and guarantees for people with HIV, as well as the Criminal Legislation on HIV/AIDS in Colombia. Next, we searched for and analyzed tutela judgments related to the violation of rights of patients with HIV. The judgments were obtained from the website of the Constitutional Court of Colombia. All these study activities were carried out in April and May 2022.

## 3. Results

### 3.1. Legislation on HIV/AIDS in Colombia 

Decree 559 of 1991 was the first legal document regulating the management of HIV/AIDS [31]. This decree comprises 78 articles, which in turn, constitute seven chapters. The motivation for the development of this decree was expressed therein, as follows: “A new communicable disease of a fatal nature has emerged, caused by the virus called human immunodeficiency virus, HIV, for which there is currently no curative treatment nor any vaccine has been developed and that, due to its particular form of transmission, constitutes a serious threat to public health”.

Decree 559 was repealed with the introduction of Decree 1543 of 1997 [32], a new legal document regulating the management of HIV, as well as AIDS and other sexually transmitted infections (STIs). Decree 1543 was introduced for five main reasons, and states that: (a) Social security is a compulsory public service and is a right, as established in the Political Constitution [33]; (b) there has been an increase in AIDS in the country, despite scientific advances; (c) an intersectoral and multidisciplinary effort is necessary to combat both HIV and AIDS; (d) the violation of the fundamental rights of HIV carriers and people suffering from AIDS has increased; and (e) taking into account the above, it is necessary to regulate the behaviors, actions, activities, and procedures for the promotion, prevention, care, and control of HIV/AIDS infection [32].

Decree 1543 of 1997 [32] comprises seventy-four articles and seven chapters. The first chapter, constituted by two articles, establishes the scope of application and concept definitions. The second chapter refers to diagnosis and comprehensive care, and includes nine articles. The third chapter, comprised of fifteen articles, deals with promotion, prevention, epidemiological surveillance, and biosafety measures. The three articles constituting the fourth chapter are concerned with aspects related to investigation. In the fifth chapter, the exercise of rights and fulfillment of duties are discussed in sixteen articles. The sixth chapter comprises nine articles that deal with organizational and coordination mechanisms. Finally, the seventh chapter includes twenty articles pertaining to procedures and sanctions. Table 1 presents a summary of Decree 1543 of 1997 [32].

Law 972 of 2005 is the first law in Colombia governing the adoption of regulations to improve the state care of populations suffering from ruinous or catastrophic diseases, especially HIV/AIDS [34]. Law 972 comprises seven articles. Comprehensive state care is considered a national priority in the fight against HIV/AIDS, including an adequate supply of drugs, reagents, and authorized medical devices for diagnosis and treatment (Art. 1). In addition, the right to life, dignity, and non-discrimination, among other fundamental rights, such as intimacy and privacy, and the right to work, family life, and study, are emphasized. Furthermore, it is stated that the essential task of health authorities is to treat and rehabilitate the patient and prevent the spread of infection (Art. 2). Laboratory, medical or hospital assistance for any person infected with HIV/AIDS is guaranteed as needed, through the public or private health system (Art. 3). In relation to the above, it is also decreed that the national government will promulgate strategies to reduce the costs of medicines, reagents, and medical devices (Art. 4), and that a centralized price and purchase negotiation system will be developed to reduce costs (Art. 5). The bodies responsible for the application of this law are the Ministry of Social Protection and departmental, district, and municipal health agencies (Art 6). Finally, it is stated that Law 972 enters into force after its sanction and publication on 15 July 2005 [34].

### 3.2. Criminal Legislation on HIV/AIDS in Colombia

HIV/AIDS has also been considered under criminal legislation in Colombia. In Law 599 of 2000 [35], in the chapter referring to crimes against public health, it is stated that “after being informed regarding infection with the HIV or hepatitis B, carrying out practices through which another person may be contaminated, such as donating blood, semen, organs or body parts, will result in a prison sentence from 3–8 years” (Art. 370).

Following this initial law, another new law that reformed the Penal Code, Law 1220 of 2008, was introduced [36]. The purpose of this law was to increase the penalties for crimes against public health. Therefore, in relation to the transmission of HIV/AIDS, the prison sentence was increased from 6 to 12 years (Art. 370).

### 3.3. Obstacles Faced by People with HIV/AIDS at the Judicial Level in Colombia

Since records began on the website of the Constitutional Court in 1992, more than 500 tutela rulings directly or indirectly related to HIV/AIDS have been issued. In these rulings, whether fundamental rights have been violated is verified, such as discrimination in the workplace, military or place of learning due to having HIV/AIDS. Whether rights to the disability pension, health, liberty, privacy (according to the principle of confidentiality) or a decent life have been violated is also verified, for both Colombian citizens and foreigners.

In the following, the five most recent Constitutional Court tutela judgments on HIV/AIDS are analyzed to understand the judicial obstacles faced by HIV/AIDS patients in Colombia.

Judgment T-077 of 2022 [37]. One man was considered to have 66.9% disability due to neurological symptoms associated with opportunistic infection by toxoplasma, with immunosuppression due to HIV infection. The affected party filed a tutela action, citing a violation and requesting the protection of his fundamental rights to human dignity, the “vital minimum“ social security, and a decent life; he considered these to have been violated by Protection S.A., a company that did not take into account the weeks of price quotation. In Judgment T-077 of 2022, the Court ruled that it grants the protection of the fundamental rights to social security and the vital minimum. In addition, it ordered the insurer (Protection S.A.) to recognize and pay the disability pension, and the National Police to recognize, liquidate, and issue the respective pension bonus or shares (as applicable), in favor of the disability pension.

Judgment T-415 of 2021 [38]. A foreign woman who was irregularly in the country was diagnosed with HIV. However, the health services denied her the right to treatment for HIV, putting her life at risk. For this reason, she filed a tutela action against the mayor’s office, as well as against the health secretary of the city in which he was located, arguing a violation of his fundamental rights to health, a decent life, physical integrity, equality and social security, and requesting the protection thereof. In Judgment T-415 of 2021, the Court ruled that it protects the fundamental rights to life and health. Therefore, the Ministry of Health was ordered to carry out a complete diagnostic workup and guarantee provision of the services necessary for establishing pathology. In addition, the interested party was ordered to complete the necessary procedure to regularize her immigration status, with the Mayor’s Office ordered to accompany the plaintiff during the process of registering with the social security system. The foregoing also applied to the children of the interested party.

Judgment T-229 of 2021 [39]. A couple filed a tutela action, claiming that their fundamental rights to due process, and access to the administration of justice, and effective judicial protection, had been violated. The husband, who was a soldier, suffered polytrauma from an antipersonnel mine explosion. As a result, he received several blood transfusions and his foot was amputated. Subsequently, and due to other health problems, he was diagnosed with HIV and the test showed that he was infected due to a blood transfusion. The husband had also infected his wife. In Judgment T-229 of 2021, the Court ruled in favor of the plaintiffs in terms of the fundamental rights to due process, access to the administration of justice, and effective judicial protection.

Judgment T-221 of 2021 [40]. A woman diagnosed with HIV, and later with tuberculosis, filed a tutela action, since the company where she worked decided not to extend her employment contract. The plaintiff requested protection of her fundamental rights to reinforced labor stability, and the right of people in conditions of manifest weakness due to illness to work, the vital minimum, life in dignified conditions, and human dignity. In Judgment T-221 of 2021, the Court ruled in favor of the plaintiff in terms of the protection of the fundamental right to labor stability, and ordered the company to reinstate the interested party in a position equal to or better than the previous one, and to pay her the wages and social benefits that she had stopped receiving.

Judgment T-031 of 2021 [41]. A man with HIV took various tests to join a company. During the selection process, he was informed of his good results, and that he would have another interview. However, after undergoing the company’s medical examinations and being informed that he had been diagnosed with HIV, the company suspended the scheduled interview and, after 2 months, informed him that his candidacy had not been successful. However, the position remained vacant. In Judgment T-031 of 2021, the Court ruled that it grants the protection of the fundamental rights to equal opportunities in relation to access to work, non-discrimination, human dignity, privacy, and due process. In addition, it condemned the company and forced them to pay moral damages to the plaintiff.

## 4. Discussion

People with HIV/AIDS not only suffer poor physical and/or psychological health, but are also victims of violations of human rights and fundamental freedoms. Although international treaties with the force of law have been ratified in Colombia [11,12,13,14,15], along with a regulatory framework to legally protect people with HIV/AIDS, it is essential to determine the effectiveness of the regulatory framework in Colombia. Therefore, the main objective of this research was to examine HIV/AIDS legislation in Colombia. To this end, decrees and laws on HIV/AIDS were analyzed, as well as the most recent tutela rulings of the Constitutional Court, to identify the main obstacles to and violations of fundamental rights. The analysis showed that legal norms have undergone modifications, and that people with HIV/AIDS have rights, as well as obligations, and may even face criminal sanctions. Finally, violations of human rights and fundamental freedoms were observed in some cases; notwithstanding, the Constitutional Court recognized these violations in the tutela judgments analyzed, and in all of them, the ruling ordered the necessary measures for the protection of the violated rights. 

After analyzing the legislation on legal assets protected by two decrees [31,32], the first law in Colombia, which was introduced to improve care by the State for those with HIV/AIDS [34], as well as two laws that modified The Penal Code [35,36], were found to have undergone a normative evolution, which is discussed below.

First, it is necessary to discuss the factors that led to the introduction of these rules. Decree 559 of 1991 was the first legal document regulating the management of HIV/AIDS in Colombia [31]. The development of Decree 559 of 1991 was motivated by the appearance of HIV/AIDS; it was articulated that, due to its deadly nature, the lack of curative treatment and vaccines, and the serious threat to public health, it was necessary to create provisions and regulations. A balance between the rights and duties of people affected by HIV/AIDS and society in general was also emphasized, and the regulation was aimed at preventing and controlling the HIV epidemic [31]. Decree 1543 of 1997 was the second legal document related to HIV/AIDS in Colombia [32]. This second document was developed due to the increase in HIV infections and AIDS among the general population, recognition of the necessity of efforts for control and prevention at the intersectoral and multidisciplinary level, and recognition of the potential for violation of the fundamental rights of people with HIV/AIDS. Therefore, the first decree [31] was developed due to the appearance of HIV/AIDS; although rights and duties were included, the main purpose was to prevent and control the pandemic. Meanwhile, the second decree [32] was developed due to the increase in infections, and since violations of the fundamental rights of people with HIV/AIDS were recognized. This represents an important action on the part of the State, where recognition of the violation of rights led to the consecration of more rights and/or duties to protect those affected. Finally, Law 972 of 2005 was developed to improve the previously proposed regulations, and the care provided by the State, in relation to HIV/AIDS [34].

Second, to understand the evolution of the regulations, it is necessary to highlight differences in the decrees [31,32]. The first difference is that Decree 1543 of 1997 [32] includes an article referring to comprehensive healthcare in different domains (prevention, diagnosis, treatment, rehabilitation, and readaptation), where the administration of drugs, which are considered effective to improve the quality of life of the infected person, is included within this care. In Decree 559 of 1991 [31], comprehensive care directed toward various domains was not considered, nor was the administration of drugs. Moreover, Decree 1543 of 1997 [32] included two other articles referring to diagnosis and comprehensive care for the first time, i.e., an article referring to updating health teams according to scientific and technological advances (Art. 10), and an article alluding to the fact that the health team should train the person responsible for (or who lives with) the patient to provide adequate care (Art. 11) [32]. Another difference between the two decrees is referred to in Chapter III. In the first decree [31], only articles referring to epidemiological prevention and control were included, while in the second decree [32], these articles were included along with additional ones referring to promotion (Art. 14 and 15) and biosecurity measures (Art. 23). The chapter on Decree 1543 of 1997 referring to research was also expanded to include new articles [32]. It is stated that experimental investigations, including people with HIV/AIDS must be approved by the Ministry of Health or other health authorized lower organisms (Art. 28, paragraph), and that in all investigations the anonymity of the participants must be guaranteed (Art. 29). In Chapter V, pertaining to the exercise of rights and fulfillment of duties, the article referring to the workplace is expanded (Art. 35 of Decree 559) [31]. Decree 1543 [32] states that the rights of workers are guaranteed in accordance with the corresponding labor legal provisions (Art. 35), and that the employer must provide job opportunities and, if necessary, relocate the person with HIV/AIDS, while maintaining their employment status (Art. 35, first paragraph). Furthermore, being infected with HIV/AIDS cannot be ground for dismissal (Art. 35, second paragraph). Another major evolution in law concerns the norms regarding the organization and coordination mechanisms. The functions of the Executive Committee are specified (Art. 51), as well as the periodicity of their meetings (Art. 52) and functions of the General Coordinator (Art. 53). Territorial Committees were also created (Art. 54).

Third, it is important to discuss laws that have modified the Penal Code in relation to HIV/AIDS. Although Law 599 of 2000 [35] stipulated imprisonment of 3–8 years for any person who, knowing they were infected with HIV/AIDS, carried out activities that could infect others, such as donating blood, semen or organs (Art. 370), Law 1220 of 2008 [36] increased the prison sentence to 6–12 years (Art. 370). With this modification, Colombia became one of the most severe countries in terms of penalizing the transmission of HIV. The Political Declaration on HIV/AIDS of the United Nations [42] and Criminalization of HIV Transmission report [43] stated that the creation of specific crimes related to HIV was not necessary, since the criminal laws already in force were deemed sufficient to punish people who intended to transmit the virus. In this sense, Art. 369 (anyone who spreads an epidemic will incur a prison sentence of 4–10 years) of the Penal Code [44] provides a pertinent legal framework. Considering the foregoing, the Colombian Constitutional Court declared the rule unnecessary and unconstitutional, and affirmed that Art. 370 of the Penal Code [36] represented an absolute restriction or annulment of the sexual rights of people carrying HIV [45]. In other words, this article constituted discrimination and violation of sexual rights and, moreover, was not effective in meeting public health objectives.

Despite the rights established in relation to regulatory norms in Colombia, people with HIV/AIDS still encounter obstacles and barriers that violate their fundamental rights. An example of this can be seen in the tutela rulings reviewed and resolved by the Constitutional Court [37,38,39,40,41]. In these five sentences, violations of fundamental rights to privacy, non-discrimination, human dignity, due process, life, and health can be observed. Typically, the plaintiffs suffered violations of fundamental rights related to the workplace [37,39,40,41] or treatment for HIV/AIDS (which was denied to a woman due to her irregular status in the country) [38]. Although, it is important to point out that the Constitutional Court recognized the violation of human rights and ruled on the necessary measures to establish the rights violated to the plaintiffs.

This research had some limitations. First, the results of the analysis of law and sentencing cannot be generalized to all people with HIV/AIDS in Colombia. Second, only the five most recent Constitutional Court rulings regarding HIV/AIDS cases were analyzed. Therefore, further investigations are recommended, in which the judgments of guardianship on HIV/AIDS should be systematically analyzed from a historical perspective, including rights violations as well as the final results of all sentences passed by the Constitutional Court.

## 5. Conclusions

HIV/AIDS is a pandemic that, despite beginning several decades ago, continues to spread and cause chronic disease, which sometimes leads to death and human rights violations of infected people. The International Conventions and Treaties on fundamental rights have been essential for establishing, promulgating, and modifying regulatory norms on HIV/AIDS. The specific decrees and laws guaranteeing the rights and duties of people with HIV/AIDS have been very important to improve the quality of life of these people, and represent progress toward addressing the obstacles and barriers that they encounter in various spheres of life. In addition, these regulations provide legal certainty and protect human rights and fundamental freedoms. Although the legal framework formally guarantees the rights of people with HIV/AIDS in Colombia, this population still suffers violations of protected rights. Among the violated rights, the following stand out: The right to health and life, the right to non-discrimination, the right to human dignity, and the right to privacy, as well as due process. The violation of these rights is confirmed in the tutela judgments analyzed, since the Constitutional Court concluded the violation of these rights and expressed the obligation to take the necessary measures to restore the violated rights. Moreover, it is important to highlight the role of the Colombian Constitutional Court, which has not only established rights through tutela rulings, but also declared an article stating that specific punishment of HIV transmission is unconstitutional. In short, despite the fact that the regulatory framework on HIV/AIDS in Colombia is adequate and guarantees the fundamental rights of those affected, infected people still suffer violations of their human rights and fundamental freedoms.

## Figures and Tables

**Table 1 ijerph-19-11423-t001:** Summary of Decree 1543 from 1997.

Chapters	Articles
Chapter 1	Art. 1. Scope of application. The provisions contemplated in this decree apply to all persons without distinction, throughout the national territory. Art. 2. Technical definitions. A set of definitions are presented (e.g., self-care, risk conditions, prevalence, etc.).
Chapter 2	Art. 3. Diagnosis. The diagnosis corresponds to an exercise in the field of medicine. Art. 4. Indications for diagnostic tests. The purpose of the diagnosis will serve for confirmation of suspected HIV; epidemiological studies; individual request of the interested person; rule out the presence of HIV. Art. 5. Performance of diagnostic tests. They will be carried out in public or private laboratories. Art. 6. Delivery of test results. The results will be delivered by medical professionals or other trained health team members to the patient. Art. 7. Person infected with HIV. It will be the asymptomatic person and therefore does not have AIDS. Art. 8. Obligation of attention. No person who provides health services can refuse to provide care to people with HIV or AIDS. Failure to provide care will result in punishable conduct. Art. 9. Comprehensive healthcare. This may be outpatient, hospital, home or community care, and will be directed toward prevention, diagnosis, treatment, rehabilitation, and adaptation. This includes drugs that are considered effective for improving the quality of life of those affected. Art. 10. Updating the health team. Public or private health entities must promote and provide information, training, and education pertaining to STDs, HIV, and AIDS, with the aim of keeping health professionals updated. Art. 11. Preparation of the family or those responsible for the patient. The health team will train the person responsible for the patient and those who live with the patient to provide adequate care.
Chapter 3	Art. 12. Promotion. This implies respect for people’s right to self-determination in terms of their sexual habits and conduct. Art. 13. Prevention. This will guarantee access to education and information; social and health services; and environmental support and tolerance based on respect for human rights. Art. 14. Intersectoriality in terms of promotion and prevention. National agencies and private health entities must carry out promotion and prevention activities. Art. 15. Protocols for comprehensive care for HIV/AIDS. The Ministry of Health will issue standards for the promotion, prevention, and care of HIV/AIDS patients, in accordance with universal scientific principles. Art. 16. Education for sexual and reproductive health. The Ministry of Education, in coordination with the Ministry of Health, will promote responsible, healthy, and ethical sexuality in children and young people. Sex education in educational institutions will be performed with the participation of the entire educational community, emphasizing the promotion of responsible attitudes and behaviors that allow the development of autonomy, self-esteem, the values of coexistence, and the preservation of sexual health; factors that contribute to the prevention of HIV/AIDS. Art. 17. Dissemination of messages. The Ministry of Communications will adopt the necessary mechanisms for targeted promotional messages to be broadcasted through the mass media to specific populations, to prevent HIV/AIDS and non-discrimination toward people with HIV/AIDS. Art. 18. Participation of NGOs. The Ministry of Health will support coordinated planning and execution of actions by NGOs, to train leaders and promote awareness of and prevent HIV/AIDS. Art. 19. Obligations of health-promoting entities (EPS). EPS are obligated to engage in promotion and prevention activities, and to provide assistance. Art. 20. Case information. The health sector must report cases of HIV/AIDS, as well as other STIs, to the Territorial Directorates of Health. Otherwise, it will be punishable conduct. Art. 21. Prohibition to perform diagnostic tests. Prohibition of the requirement for testing to access educational, sports, social or rehabilitation centers; to access the workforce and achieve a permanent place therein; to enter or reside in the country; to access health services; enter or carry out cultural, social, political, economic or religious activities. Art. 22. Carrying out tests in blood and organ banks. Blood and organ donors should be tested for HIV. Art. 23. Biosafety. Public and private healthcare entities, laboratories, blood banks, and other related entities must take universal security measures into account. Art. 24. Availability of condoms. Pharmacies, drugstores, and other outlets must guarantee the availability of condoms. Art 25. Prohibition of requirement for identity cards. The requirement of an identity card or certificate with reference to HIV/AIDS or other STIs is prohibited. Art. 26. Surveillance of NGOs. NGOs providing services related to HIV/AIDS will be subject to assessment and surveillance.
Chapter 4	Art. 27. Norms for therapeutic research. Subject to the Declaration of Helsinki, until specific internal legal provisions are issued. Art. 28. Incentive for research. The Ministry of Health will promote research on HIV/AIDS. Art. 29. Epidemiological investigation. The anonymity of individuals must be guaranteed.
Chapter 5	Art. 30. Duties of the community. People should practice the necessary self-care and prevention measures to prevent the infection and spread of HIV. Art. 31. Duties of health entities and health team members. Obligation to provide comprehensive care for people with HIV/AIDS. Art. 32. Duty of confidentiality. Medical professionals must maintain the confidentiality of people with HIV/AIDS. Art. 33. Clinical history. Private document whereby, to be known by third parties, authorization from the owner or according to law is necessary. Art. 34. Disclosure of professional secrecy. For health reasons, the doctor may reveal information to an infected person; relatives (if pertinent to treatment); those responsible for the infected person (if they are minors or mentally incompetent); interested parties in danger of infection, the spouse, sexual partner or offspring; and judicial or health authorities if required by law. Art. 35. Employment status. Workers are not required to tell their employers if they have HIV/AIDS. The rights of workers will be guaranteed in accordance with the corresponding labor legal provisions. In addition, the employer must relocate the person with HIV/AIDS if necessary and being infected cannot be cause for dismissal. Art. 36. Duty to inform. The person with HIV/AIDS must inform their sexual partner and the doctor or health team from whom they request assistance to ensure proper treatment and prevent the spread of the virus. Art. 37. Right to informed consent. Informed consent will be required to conduct prevalence assessments or surveys. Art. 38. Deprivation of liberty. Prisoners cannot be forced to undergo HIV testing. Art. 39. Right to non-discrimination. Persons with HIV/AIDS, their children, and relatives, may not be denied entry or permanence in educational, public or private, assistance or rehabilitation centers, nor access to any work activity or permanence therein, for this reason, nor will they be discriminated against for any reason. Art. 40. Prohibition of diagnostic test requirements for access to services. Health entities may not require HIV diagnostic tests for access to protection and health services. Art. 41. Duty not to infect. The person with HIV must refrain from donating blood, semen or organs, as well as from carrying out activities that may infect other people. Art. 42. Right to sexual promotion, prevention, and education. All citizens have this right. Art. 43. Right to information on health status. All citizens have this right. Art. 44. Right to die with dignity. All citizens have this right. Art. 45. Burial or cremation. People with HIV have this right.
Chapter 6	Art. 46. Organizational structure. Institutions and coordination mechanisms are to be established in relation to HIV/AIDS, which are established in the following articles. Art. 47. National AIDS Council. This council is intended to be a permanent adjunct to the Ministry of Health and will be composed of: (a) Minister of Health or his Vice Minister, who will preside over it; (b) Minister of Education or its Vice Minister; (c) Minister of Communications or its Deputy Minister; (d) Minister of Labor or his Deputy Minister; (e) Ombudsman or his delegate; (f) Director of the Colombian Institute of Family Welfare or his delegate; (g) Director of the National Institute of Health; (h) Director of the National Institute for Food and Drug Surveillance; (i) Head of the National Program for the Prevention and Control of STDs and AIDS; (j). Coordinator of the National Project of Sexual Education of the Ministry of Education; (k) Delegate of the UNAIDS Theme Group for Colombia; (l) Two legally constituted representatives of NGOs fighting AIDS, appointed by the National Coordinating Table of NGOs; (m) Man and woman infected with HIV, representatives of the Support and Self-support Groups, chosen by the Ministry of Health; (n) Representative of the Health Promoting Entities; (o) Representative of the National Television Commission; (p) Representative of the National Network of Blood Banks; (q) Representative of the Intersectoral, Departmental, and Municipal Committees chosen by the Ministry of Health. Art.48. Functions. The National AIDS Council will develop and implement functions, such as: Proposing general policies for the promotion, prevention, and care of HIV/AIDS; recommend mechanisms to promote intersectoral participation; evaluate the development of the National Program for the Prevention and Control of STDs and AIDS; work toward securing financial resources; provide advice to aid the development of international projects, agreements or conventions; approve coordination mechanisms with other countries; draft its own regulations. Art. 49. Meetings. The National AIDS Council will meet every 6 months or more frequently in certain circumstances. Art. 50. Executive Committee. This committee will be created specifically in relation to HIV/AIDS and STIs and will be composed of (a) The General Director of Promotion and Prevention of the Ministry of Health; (b) The Head of the National Program for the Prevention and Control of STDs and AIDS; (c) The Head of Health Education of the Ministry of Health; (d) The Head of Sexual and Reproductive Health of the Human Development program of the Ministry of Health; and (e) The Head of Priority Actions in Health of the Ministry of Health. Art. 51. Functions. The Executive Committee will direct the development of promotion, prevention, assistance, and control strategies; promote the participation of NGOs, people with HIV/AIDS, and other stakeholders; adopt UNAIDS recommendations; develop, direct, and evaluate the implementation of the National Program for the Prevention and Control of STIs and AIDS; present and deliver reports prior to the meetings of the National AIDS Council; regulate intersectoral committees; draft its own regulations. Art. 52. Meetings. The Executive Committee will meet every 2 months and at the request of its members. Art. 53. Functions of the General Coordinator. Fundamentally, the General Coordinator will convene and coordinate the activities of the Executive Committee. Art. 54. Creation of territorial committees. Intersectoral committees are to be created on a permanent basis at the Sectional, District, and Local levels in relation to HIV/AIDS.
Chapter 7	Art. 55. Spread of the epidemic. Failure to comply with articles 36 and 41 of this decree may lead to a complaint. Art. 56. Sanctions. Failure to comply with this decree may lead to sanctions: Fines in the amount of up to 200 monthly legal minimum wages; Intervention of the administrative and/or technical management of the entities that provide health services for up to a term of 6 months; Suspension or definitive loss of legal status of private persons who provide health services; Suspension or loss of authorization for the provision of health services. Art. 57. Penalty process. This process can be started by anyone. Art. 58. Provision of information to the competent authority. If the facts of the sanctioning process constitute a crime, the competent authority will be informed. Art. 59. Complainant. You can intervene in the course of the procedure. Art. 60. Existence of another process. The existence of another criminal process will not suspend the sanctioning process. Art. 61. Research. Once the complaint is received, investigation of the facts will begin. Art. 62. Cessation of procedure. This will occur when there is no evidence to support the investigation. Art. 63. Formulation of charges. Once the investigations have been carried out, the offender will be charged. Art. 64. Notification. Notification will be performed in person or by certified mail. Art. 65. Releases. Once the notification is received, the offender will have 10 days to present their defense. Art. 66. Tests. The competent authority will decree the appropriate tests within a maximum of 30 days. Art. 67. Sanctions. Within 10 days of the term of the previous article expiring, the health authority will qualify the fault and impose the corresponding sanction. Art. 68. Aggravating circumstances. These include acting with ignoble motives; carrying out the damaging act with the complicity of subordinate persons; failure to protect another; eschewing responsibility; taking advantage of a calamitous or dangerous situation; and premeditating the offence. Art. 69. Mitigating circumstances. Good background; lack of clarity when it has influenced the fact; and attempts to repair the damage. Art. 70. Exemption from liability. There is no liability when there has been no violation of the rules. Art. 71. Notification of sanctions. Personal notification or via edict. Art. 72. Sanctioning authority. These will include the Territorial Sectional, District or Local Directorate of Health. Art. 73. Validity of the sanction. The proposal itself. Art. 74. Validity. This decree repeals Decree 559 of 1991.

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
