# Peer review of "Protection of Human Rights and Barriers for People with HIV/AIDS in Colombia: An Analysis of the Legal Framework"

_ijerph, 2022, doi:10.3390/ijerph191811423_

Round 1

Reviewer 1 Report

This article explores a very significant concern regarding the rights of persons with HIV.  It is well written and provides a good overview of the issue in Colombia.  Some points to consider to improve the article:

In what ways and to what degree is Colombian law influenced by international human rights law?

Were the court decisions in any way specially based in the international human rights law?

The analysis of the court decisions suggests that in all cases the rights of the plaintiffs were upheld, yet the authors conclude that rights were violated.  There seems to be an inconsistency between the analysis and the conclusions drawn.  Please clarify.

At least to this reviewer, the purpose of the paper remains somewhat unclear:  is the purpose to provide more a summary or is it to provide a legal analysis?  If the latter, what is the legal theory being advanced?

Author Response

Comments Reviewer 1

Comment. This article explores a very significant concern regarding the rights of persons with HIV.  It is well written and provides a good overview of the issue in Colombia. Some points to consider to improve the article:

Response: The authors appreciate your review, as well as your suggestions that undoubtedly improve our paper.

Comment. In what ways and to what degree is Colombian law influenced by international human rights law?

Response: Colombian law is influenced by international human rights law, first, because human rights are universal; second, because the Colombian State has ratified international treaties. An example of this is the Convention against Torture and Other Cruel, Inhuman or Degrading Treatment or Punishment. This Convention was ratified by Colombia on December 10, 1986. It was through Law 70 of 1986 (of December 15, 1986) through which the "Convention against Torture and Other Cruel, Inhuman or Degrading Treatment or Punishment" was approved, adopted at the United Nations on December 10 from 1984. Another example is Law 248 of 1995 through which the Inter-American Convention to prevent, punish and eradicate violence against women was ratified, (Congress of the Republic, Colombia), of December 29, 1995. Therefore, we make the following statement (lines 90-97) “In accordance with the various binding instruments, Colombia, among other countries, must respect, protect and comply with the following. First, they must respect human rights norms, and thus must not directly or indirectly violate the fundamental rights of people living with HIV/AIDS. Second, they must protect and adopt the necessary measures to prevent the human rights and fundamental freedoms of people living with HIV/AIDS from being violated. Third, they must comply with the established norms, and develop and implement all necessary legislative, financial, administrative and judicial measures”. In addition, we inform in the first paragraph of the discussion that Colombia has ratified international treaties “Although international treaties with the force of law have been ratified in Colombia [11-15]”.

Comment. Were the court decisions in any way specially based in the international human rights law?

Response: Yes, the judgments were based on international law, since the fundamental pillars of the Political Constitution are the international treaties on human rights. Tutelage action can be defined as: The action for tutelage against court rulings is an exceptional instrument, aimed at dealing with those situations in which the judge's decision incurs serious flaws, of constitutional relevance, which make the decision incompatible with the Constitution”.  In addition, in the tutela judgments analyzed, it was found that the Constitutional Court recognized the violation of human rights and fundamental freedoms and for this reason the sentences ruled in favor of the protection of human rights. The authors have included the following clarification at the end of the first paragraph of the discussion: “notwithstanding, the Constitutional Court recognized these violations in the tutela judgments analyzed and in all of them the ruling ordered the necessary measures for the protection of the violated rights”.

Comment. The analysis of the court decisions suggests that in all cases the rights of the plaintiffs were upheld, yet the authors conclude that rights were violated.  There seems to be an inconsistency between the analysis and the conclusions drawn. 

Response: Done, we have made a clarification in the text (lines 337-339): “Although, it is important to point out that the Constitutional Court recognized the violation of human rights and ruled on the necessary measures to establish the rights violated to the plaintiffs”.

Comment. At least to this reviewer, the purpose of the paper remains somewhat unclear:  is the purpose to provide more a summary or is it to provide a legal analysis?  If the latter, what is the legal theory being advanced?

Response. The main purpose is to examine the effectiveness of the regulatory framework and for this we carry out, on the one hand, a review of the decrees and laws on HIV/AIDS and, on the other hand, we analyze the tutela judgments to verify which rights were violated. So, in response to your question, more than a legal theory, it can be said that the information is extracted literally when reading the sentences to know the rights violated.

Reviewer 2 Report

The manuscript consists of total 12 pages, including the list of total 46 literature references. The title corresponds adequately wit the contents of the manuscript. The article summarizes the historical development and current status of legal regulations of the status of HIV-infected people in Columbia. As such, it fits into the scope of works published in the Journal. The manuscript is written in English of satisfactory quality and has a logical structure.

I have a bit mixed feelings about the text. The task accepted by the Authors is very ambitious, as it is difficult to present a topic as wide and deep as legal situation of HIV-infected people, including both the international and local ramifications, in a merely 12-pages long text and at the same time not make on the Reader the impression of only barely superficial and quite general insight, as every Reader will have his/her own interests in some more detail in some aspect of the problem, which obviously could not have been fitted in in the desired extent. In such a situation, the Authors' role may be rather not to dive into details but rather to sketch a legal regulations framework for the Readers to be followed further independently and the Authors apparently have chosen this path. However, in order to succeed the Authors must include in case of each of the referred sources a direct Internet link to the full text of the given regulation, including the most basic international law regulations, so the Readers are able use the provided framework efficiently and with maximal ease.

The Authors decided to provide also a summary of the legal status, along with some legal cases, which is interesting. However, the legal regulations presentation, especially the table 1 listing 74 articles of a regulation, shall be reworked. In its current form, it is of little use for a Reader not intending/or unable/ to read the full given local regulation, as the information provided, especially in the table, is too general and only includes header-like information on what is regulated, but in most cases without any actual information on HOW actually the different aspects are regulated in Colombia - while that naturally is the most interesting aspect for the Readers they would seek in the article, especially as the most of them are not able to read the Colombian regulation in its native language.  

The Abstract is too general, in particular it does not include any concrete information or conclusion resulting from the vast review performed by the Authors.

The Introduction provides enough background for the investigated problem.

The Material and methods are presented in enough detail.

The Results need to be changed into more concrete form as suggested above.

The Discussion is based on the presented results.

The Conclusions are in part not based firmly enough on the discussed results - the Authors provide the data on rights and their legal regulations framework but there is not enough basis in the presented material to point towards the most commonly violated ones.

The table 1 shall be changed into more concrete from as suggested above.

The literature references are numerous and recent enough, relevant to the topic; direct Internet links need to be added to all legal regulations so the Readers are able to use directly the framework provided by the Authors.

Author Response

Comments Reviewer 2

Comment. The manuscript consists of total 12 pages, including the list of total 46 literature references. The title corresponds adequately with the contents of the manuscript. The article summarizes the historical development and current status of legal regulations of the status of HIV-infected people in Columbia. As such, it fits into the scope of works published in the Journal. The manuscript is written in English of satisfactory quality and has a logical structure.

Response: The authors appreciate your review, as well as your suggestions that undoubtedly improve our paper.

Comment. I have a bit mixed feelings about the text. The task accepted by the Authors is very ambitious, as it is difficult to present a topic as wide and deep as legal situation of HIV-infected people, including both the international and local ramifications, in a merely 12-pages long text and at the same time not make on the Reader the impression of only barely superficial and quite general insight, as every Reader will have his/her own interests in some more detail in some aspect of the problem, which obviously could not have been fitted in in the desired extent. In such a situation, the Authors' role may be rather not to dive into details but rather to sketch a legal regulations framework for the Readers to be followed further independently and the Authors apparently have chosen this path. However, in order to succeed the Authors must include in case of each of the referred sources a direct Internet link to the full text of the given regulation, including the most basic international law regulations, so the Readers are able use the provided framework efficiently and with maximal ease.

Response. Done. The authors have included a direct link to the full text of the mentioned regulations. The link has been included in the bibliographical references.

Comment. The Authors decided to provide also a summary of the legal status, along with some legal cases, which is interesting. However, the legal regulations presentation, especially the table 1 listing 74 articles of a regulation, shall be reworked. In its current form, it is of little use for a Reader not intending/or unable/ to read the full given local regulation, as the information provided, especially in the table, is too general and only includes header-like information on what is regulated, but in most cases without any actual information on HOW actually the different aspects are regulated in Colombia - while that naturally is the most interesting aspect for the Readers they would seek in the article, especially as the most of them are not able to read the Colombian regulation in its native language.  

Response. The authors have included more information in some of the articles in the table. However, Decree 1543 of 1997 regulates the management of HIV/AIDS but does not specify the “how”. For example, the articles referring to promotion and prevention are general in the standard itself, since no more information is specified than that provided in the articles. We have included more information in the articles included in chapter VI on organization and coordination mechanisms. It is important to clarify that the decree is part of what in law is called "Substantial Law", that is, they are conceptual legal norms that define values, principles and guiding norms to make decisions. On the other hand, there is "Procedural Law", which are those other legal norms that structure procedures, guides to follow or step by step to obtain substantial rights. That is, procedural law deals with the "how."

Comment. The Abstract is too general, in particular it does not include any concrete information or conclusion resulting from the vast review performed by the Authors.

Response. The authors have modified the abstract and added the main results and conclusions. We have added the following: “It is verified that there is a specific regulation on HIV/AIDS, specifically decree 559 of 19991, decree 1543 of 1997, Law 599 of 2000, Law 972 of 2005 and Law 1220 of 2008. Although at the legislative level Colombia shows an evolution in the norm, patients with HIV/AIDS continue to be victims of human rights violations. Thus, and through the analysis of tutela judgments, it was found that the Constitutional Court recognized the violation of rights and ordered the necessary measures to be taken to guarantee the human rights and fundamental freedoms of the defendants”.

Comment. The Introduction provides enough background for the investigated problem.

Response: Thank you for your comment.

Comment. The Material and methods are presented in enough detail.

Response: Thank you for your comment.

Comment. The Results need to be changed into more concrete form as suggested above.

Response: We have included more information in some table items. However, take into account the difference between substantive law and procedural law, previously explained.

Comment. The Discussion is based on the presented results.

Response: Thank you for your comment.

Comment. The Conclusions are in part not based firmly enough on the discussed results - the Authors provide the data on rights and their legal regulations framework but there is not enough basis in the presented material to point towards the most commonly violated ones.

Response: Done. We have included the following clarification: “The violation of these rights is confirmed in the tutela judgments analyzed, since the Constitutional Court concluded the violation of these rights and expressed the obligation to take the necessary measures to restore the violated rights”.

Comment. The table 1 shall be changed into more concrete from as suggested above.

Response: We have included additional information in some articles. However, take into account the difference between substantive law and procedural law, previously explained.

Comment. The literature references are numerous and recent enough, relevant to the topic; direct Internet links need to be added to all legal regulations so the Readers are able to use directly the framework provided by the Authors.

Response. Done. We have included the links. Thank you for your comment.

Round 2

Reviewer 1 Report

I believe the authors' responses are good and improve the article.